# The RNA Polymerase Inhibitor Corallopyronin A Has a Lower Frequency of Resistance Than Rifampicin in *Staphylococcus aureus*

**DOI:** 10.3390/antibiotics11070920

**Published:** 2022-07-08

**Authors:** Jan Balansky, Kenneth Pfarr, Christiane Szekat, Stefan Kehraus, Tilman Aden, Miriam Grosse, Rolf Jansen, Thomas Hesterkamp, Andrea Schiefer, Gabriele M. König, Marc Stadler, Achim Hoerauf, Gabriele Bierbaum

**Affiliations:** 1Institute for Medical Microbiology, Immunology and Parasitology, University Hospital Bonn, 53127 Bonn, Germany; s6jabala@uni-bonn.de (J.B.); kenneth.pfarr@ukbonn.de (K.P.); christiane.szekat@ukbonn.de (C.S.); tilman.aden@ukbonn.de (T.A.); andrea.schiefer@uni-bonn.de (A.S.); hoerauf@uni-bonn.de (A.H.); 2German Center for Infection Research (DZIF), Partner Site Bonn-Cologne, 53127 Bonn, Germany; 3Institute of Pharmaceutical Biology, University of Bonn, 53115 Bonn, Germany; kehraus@uni-bonn.de (S.K.); g.koenig@uni-bonn.de (G.M.K.); 4Department of Microbial Drugs, Helmholtz Centre for Infection Research, 38124 Braunschweig, Germany; miriam.grosse@helmholtz-hzi.de (M.G.); rolf.jansen@helmholtz-hzi.de (R.J.); marc.stadler@helmholtz-hzi.de (M.S.); 5German Center for Infection Research (DZIF), Partner Site Hannover-Braunschweig, 38124 Braunschweig, Germany; 6Translational Project Management Office (TPMO), German Center for Infection Research (DZIF), 38124 Braunschweig, Germany; thomas.hesterkamp@helmholtz-hzi.de

**Keywords:** Corallopyronin A, *Staphylococcus aureus*, rifampicin, mutation frequency, mutation rate, natural product, MRSA, VISA

## Abstract

Corallopyronin A (CorA) is active against Gram-positive bacteria and targets the switch region of RNA polymerase. Because of the high frequency of mutation (FoM) leading to rifampicin resistance, we determined the CorA FoM in *S. aureus* using fluctuation analysis at 4 × minimum inhibitory concentration (MIC). Resistant mutants were characterized. *S. aureus* strains HG001, Mu50, N315, and USA300 had an MIC of 0.25 mg/L. The median FoM for CorA resistance was 1.5 × 10^−8^, 4.5-fold lower than the median FoM of 6.7 × 10^−8^ for rifampicin, and was reflected in a 4-fold lower mutation rate for CorA than rifampicin (6 × 10^−9^ for CorA vs. 2.5 × 10^−8^ for rifampicin). In CorA-resistant/rifampicin-sensitive strains, the majority of amino acid exchanges were S1127L in RpoB or K334N in RpoC. *S. aureus* Mu50, a rifampicin-resistant clinical isolate, yielded two further exchanges targeting amino acids L1131 and E1048 of the RpoB subunit. The plating of >10^11^ cells on agar containing a combination of 4 × MIC of rifampicin and 4 × MIC of CorA did not yield any growth. In conclusion, with proper usage, e.g., in combination therapy and good antibiotic stewardship, CorA is a potential antibiotic for treating *S. aureus* infections.

## 1. Introduction

Corallopyronin A (CorA) is an α-pyrone antibiotic (MW 527.7 Da) produced by the myxobacterium *Corallococcus coralloides* Cc c127 [1]. CorA is active against Gram-positive organisms including *Staphylococcus aureus* and some Gram-negatives such as *E. coli* DH21tolC, a mutant strain with increased permeability of the outer membrane; *Neisseria flava* [1]; and the intracellular bacteria *Chlamydia trachomatis* [2], *Wolbachia* spp. [3], *Rickettsia typhi*, and *Orientia tsutsugamushi* [4]. CorA is in preclinical development for the anti-wolbachial treatment of filarial diseases [5]. 

CorA inhibits bacterial DNA-dependent RNA polymerase (RNAP) by blocking the switch region and thus prevents the opening and closing of the RNA clamp [6]. CorA binds to residues in the β (RpoB) and β’ (RpoC) subunits of RNAP. In a saturated mutagenesis experiment with the related pyrone myxopyronin and *E. coli*, exchanges of two amino acids in RpoC (K345 and V1351) and four amino acids in RpoB (V1275, E1279, S1322 and L1326) increased the CorA MIC at least eightfold [7]. Furthermore, the crystal structure of a derivative of the related compound myxopyronin bound to the RNAP of *Thermus thermophilus* showed that of these sites, K345, E1279, and S1322 directly interact with myxopyronin [6]. The CorA and myxopyronin-resistance-conferring mutations are not located in the active site of RNAP and do not confer cross-resistance to rifampicin [7]. 

After the selection of resistant *S. aureus* SH1000 mutants with CorA, five mutated amino acids were identified: S1127, E1084, and L1131 in *S. aureus* RpoB and K334 and L1165 in *S. aureus* RpoC [8]. Four of these residues directly correspond to the above-mentioned residues in *E. coli* (S1322, E1279, and L1326 in *E. coli* RpoB and K345 in *E. coli* RpoC) [8]. *Staphylococcus aureus* L1165 is located next to K1166 (K1348 in *E. coli* RpoC), which also shows an interaction with myxopyronin [6]. Additionally, this publication reported that the frequency of mutation (FoM) for resistance to CorA is high, and about equal to that of rifampicin. However, the MIC values for CorA (2 mg/L) reported in that study were higher than those published previously [1,9]. Today, preparations of CorA with increased purity are available from a recently developed heterologous producer strain [10]. In addition, CorA tends to isomerize when stored at room temperature and loses activity in stored agar plates [11]. Therefore, we chose to repeat the investigation of the mutation frequency with a >91% pure preparation of CorA, fresh agar plates, and *S. aureus* HG001, a closely related strain that, however, does not harbor the UV-treatment-induced SNPs of *S. aureus* SH1000 [12]. In this study, we show that the FoM of CorA is indeed lower than that of rifampicin and present resistance mutations obtained with different *S. aureus* clinical strains. 

## 2. Results

### 2.1. Stability of CorA in Agar Plates

Measured mutation rates are influenced by the concentration of the selective agent in the agar [13]. In a preliminary plating experiment, an agar plate with 0.5 mg/mL CorA (2 × MIC) showed more resistant colonies than agar plates that contained 1 mg/mL CorA or 2 mg/mL, indicating that the CorA concentration in the agar plate has an influence on the FoM. Therefore, before plating bacteria, the stability of CorA was tested using GC agar (Becton-Dickinson, Heidelberg, Germany). At least 75% CorA was still present after 48 h of incubation at 37 °C (Appendix A). Longer incubation times resulted in a decrease in the CorA content (Appendix A). Therefore, agar plates were prepared fresh before plating, dried for only a short time, and stored at 4 °C for a maximum of a week. In addition, CorA is sensitive to light and, therefore, the agar plates were always stored in the dark and protected from light when handled.

### 2.2. Determination of Mutation Frequency

CorA had an MIC of 0.25 mg/l against *S. aureus* strains HG001, Mu50, N315, and USA300 (Table 1). To determine the spontaneous FoM (resistant cells per total cells in the culture) leading to CorA resistance, forty parallel cultures were assayed on four different days on agar plates containing 1 mg/L CorA (4 × MIC). The FoM leading to resistance to rifampicin (MIC vs. *S. aureus* HG001, 0.0019 mg/L) was determined in parallel, using rifampicin (4 × MIC, 0.0076 mg/L) for the preparation of selective plates. Employing directly plated 100 µL culture aliquots, the percentage of agar plates that did not show the growth of resistant colonies was significantly higher for CorA (37.5%) than for rifampicin (2.9%) (*p* = 0.004, Fisher’s Exact Test). Therefore, 250 or 1000 µL of culture was pelleted, resuspended in 100 µL of buffer, and plated. Using this method, the median FoM to rifampicin resistance was 4.5-fold higher than that to CorA resistance (Table 2) and the differences between mutation frequencies were statistically significant (*p* < 0.0001, Mann–Whitney U-test) (Figure 1A; raw data sets are shown in Appendix A).

The mutation rates (Table 2) were 2.8- and 4.2-fold higher for rifampicin than for CorA, according to the Lea–Coulson Method of the Median and the Ma–Sandri–Sarkar Maximum Likelihood Estimation (Figure 1B), respectively, and were also significantly different (*p* < 0.0001, Mann–Whitney U-test). The plating of 1.2 × 10^11^ cells of *S. aureus* HG001 on Iso-Sensitest agar containing a combination of both agents, 4 × MIC of rifampicin and 4 × MIC of CorA, did not yield any growth after 24 h of incubation.

### 2.3. Mutations in CorA Resistant Strains

Twenty-seven resistant mutants from nine independent cultures of *S. aureus* HG001 were sequenced and were found to harbor exchanges in two mutational hotspots, i.e., position 1127 in RpoB (favoring the exchange S1127L) and position 334 in RpoC (favoring the exchange K334N) (Figure 2). Rifampicin-resistant strains have mutations in *rpoB*, which might influence the sites available for mutations to CorA resistance and the degree of resistance that can be reached. Therefore, resistance to CorA in a rifampicin-resistant background was tested using eight independent cultures of *S. aureus* Mu50, a clinical isolate that is resistant to methicillin and rifampicin and intermediately resistant to vancomycin. Eleven CorA-resistant colonies were sequenced. Here, additional exchanges of amino acids in RpoB at position L1131 and E1048 were found. Mutations were also sequenced in the clinically relevant, rifampicin-susceptible strains, *S. aureus* N315 and *S. aureus* USA300 NRS384, with exchanges in the same hotspots as in *S. aureus* HG001 (S1127 in RpoB and in K334 in RpoC). All sites that were affected in different strains are shown in Figure 2. The MICs of the clinical wildtype strains (Mu50, N315, and USA300 NRS384) were 0.25 mg/L CorA, whereas the MICs of resistant mutants ranged from 4 to 128 mg/L (Table 1).

## 3. Discussion

In a previous publication, FoM values of 7.2 × 10^−8^ ± 2.4 × 10^−8^ for resistance to CorA and of 1.4 × 10^−7^ ± 2.2 × 10^−7^ for resistance to rifampicin at 4 × MIC were determined, and the authors concluded that both FoMs were “comparable”. However, the exact methodologies of the experiment and calculations were not described [8]. In the current study, we repeated these experiments and took care to use agar plates with fresh CorA, because, as a neat active pharmaceutical ingredient, CorA is unstable and tends to isomerize at room temperature [11]. This resulted in significant differences in the median FoMs. The mutation rates of resistance to CorA were, depending on the method of calculation, 2.8-4.2-fold lower than those of resistance to rifampicin. In addition, 30% of the CorA plates that had received only 0.1 mL inoculum showed no growth, which was significantly higher than that for rifampicin plates. For the related antibiotic myxopyronin, mutation rates of 6 × 10^−8^ (calculated using the MSS-MLE method) were reported at 4 × MIC, which is higher than the rates of 6 × 10^−9^ to 9 × 10^−9^ obtained here using the same calculation method. Potentially unstable features have also been identified in myxopyronin. However, chemical modifications of these functional groups led to decreases in antibacterial activity [15,16].

The spontaneous mutation rates of growing bacterial cultures are strain-specific and constant in the absence of agents that influence the mutation rates. Therefore, the lower FoM and mutation rates measured with CorA might depend on a lower number of different mutations that convey resistance against CorA compared to rifampicin. For rifampicin, 16 different exchanges have been described in *S. aureus* 8325-4 after the plating of saturated cultures on agar containing 4 × MIC, and further mutations have been found in clinical isolates [17]. Saturation mutagenesis and a subsequent selection of mutants with CorA yielded seven different exchanges in six sites in *E. coli*. However, using myxopyronin, nine mutations were detected, which also increased CorA MICs eightfold or more [7]. In this study, the sequencing of 27 resistant mutants of *S. aureus* HG001 picked from nine different cultures yielded only four different exchanges targeting mainly two amino acids, S1127 in RpoB and K334 in RpoC, with S1127L RpoB and K334N RpoC being the most abundant exchanges (Figure 2). Identical exchanges were also present in the clinical strain and the community-associated MRSA (*S. aureus* N315 and *S. aureus* NRS384, respectively). We also tested the mutational changes in a rifampicin-resistant strain, *S. aureus* Mu50, an MRSA with intermediate susceptibility to vancomycin [18] that already harbors the H481Y exchange in RpoB [19]. Eleven sequenced resistant *S. aureus* Mu50 yielded two additional mutation sites at L1131 and E1084 in RpoB. Mariner et al. found exchanges in the same positions in eight sequenced mutants, including an additional exchange in L1165 [8]. In a similar study with myxopyronin, 11 different mutations from 17 independent mutants and a higher mutation rate in the range of rifampicin were reported [13]. Six of these eleven mutants, however, exhibited relatively low resistance levels (20 × MIC of the wildtype), comparable to that of the *S. aureus* Mu50 CSz1 mutant in this study. Therefore, it is possible that in these experiments relatively more mutants with a lower resistance level were able to survive the selection [13]. 

In conclusion, six possible amino acid positions were identified that yield resistance against CorA in *S. aureus* in direct selection experiments, which might explain the lower mutation rates compared to rifampicin. A stable formulation is now available [11]; therefore, with proper usage, e.g., in combination therapy and good antibiotic stewardship, CorA is a potential antibiotic for treating *S. aureus* infections. The efficacy of CorA as a chemotherapeutic agent in combination with different antibiotics for the treatment of staphylococcal disease will be investigated in the near future.

## 4. Materials and Methods

### 4.1. Strains

*Staphylococcus aureus* HG001, a derivative of the standard strain *S. aureus* NCTC 8325 (clonal complex (CC) 8) that has been repaired in *rsbU* [12], was used for the determination of mutation frequency and the characterization of mutants. *S. aureus* Mu50 (ATCC700699, rifampicin-resistant, methicillin-resistant (MRSA), vancomycin-intermediate susceptibility, CC5 [18]) was employed to test the influence of a rifampicin-resistant background on the generation of amino acid exchanges in RpoB and RpoC. The MICs and mutations of *S. aureus* USA300 (NRS384) (community-associated MRSA, source NARSA, CC8) and *S. aureus* N315 (NRS70, MRSA, rifampicin-susceptible, CC5 [18]) were also determined. Mutant strains are available from the authors.

### 4.2. Corallopyronin

CorA with a purity of >91% was produced by the Helmholtz Centre for Infection Research, Department of Microbial Drugs, using the heterologous producer strain *Myxococcus xanthus* carrying the CorA biosynthesis gene cluster, and was purified as reported previously [10]. 

### 4.3. Minimal Inhibitory Concentration

The determination of MICs was performed in Iso-Sensitest broth (Oxoid, Thermo Scientific, Waltham, MA, USA) with an inoculum of 1–5 × 10^5^ cells per ml, as specified by EUCAST [20]. Rifampicin and CorA stock solutions in DMSO (dimethyl sulfoxide) were stored in the dark at −20 °C. DMSO stock solutions were diluted with broth directly before the preparation of microtiter plates (1 µg/mL for rifampicin, 8 µg/mL for CorA as the final concentration in the stock solutions for susceptible strains and up to 512 µg/mL CorA for resistant strains). A solvent control of 12.8% of DSMO was included. Plates were read after 24 h of incubation at 37 °C.

### 4.4. Determination of Mutation Frequency and Mutation Rates

Iso-Sensitest agar (Oxoid, Thermo Scientific, Waltham, MA, USA) was prepared as advised by the manufacturer, autoclaved, and equilibrated at 50 °C before the addition of antibiotics (final concentration: 4 × MIC) for the preparation of selective agar plates. Agar plates (20 mL) were poured shortly before plating, wrapped in aluminum foil, and stored at 4 °C and never longer than for one week. All steps that involved the handling of selective agar plates were performed in the absence of artificial light, since both antibiotics are sensitive to light. For preculture, 5 mL broth were inoculated with one colony of the test strain *S. aureus* HG0001 from a fresh Columbia blood agar plate and incubated overnight. In the morning, 9–15 main cultures (9.9 mL Iso-Sensitest broth in a 100 mL Erlenmeyer flask) were inoculated with 0.1 mL preculture and incubated at 37 °C and 170 rpm until the cultures had reached an optical density OD_600_ of 1. Then, 1 mL of a culture of OD 1 (or 0.25 mL for most agar plates with rifampicin) was centrifuged (8000 rpm, 3 min) and the pellets were resuspended in 100 µL of broth. An aliquot of 100 µL of this resuspended pellet and 100 µL of undiluted culture were plated onto selective plates with CorA (1 µg/mL) and rifampicin (0.076 µg/mL). In addition, serial tenfold dilutions of the cultures were plated onto Iso-Sensitest agar without antibiotics for the determination of colony forming units (CFU). CFU were counted after 24 h of incubation at 37 °C in the dark. All culture plates that did not yield colony smears were evaluated, i.e., for CorA 26 of 40 agar plates with plated pellets were evaluated. For rifampicin, 34 main cultures were tested and it was possible to evaluate 19 of 34 plated pellets.

A two-tailed Mann–Whitney-U-Test was used for the statistical comparison of the median mutation frequencies and median mutation rates. Fisher’s exact test was employed for the comparison of agar plates without growth (GraphPad Prims version 9.2 for Windows, GraphPad Prism Software, San Diego, CA, USA). Mutation rates were analyzed using the FALCOR Fluctuation AnaLysis CalculatOR (https://lianglab.brocku.ca/FALCOR/, accessed on 18 March 2022) [14].

For the determination of FoM against a combination of both agents, rifampicin and CorA, fresh agar plates with 4 × MIC of both antibiotics were prepared two days before plating. Eight different overnight cultures were grown from eight different colonies of *S. aureus* HG001 in 5 mL of Iso-Sensitest broth. A total of 4.9 mL of culture was pelleted, and the cells were resuspended in 500 µL of Iso-Sensitest broth. These suspensions were plated in 150 µL aliquots on agar plates containing the combination of both antibiotics. For the determination of the total cell counts, a series of tenfold dilutions was plated on Iso-Sensitest agar without antibiotics. All agar plates were incubated at 37 °C and evaluated after 24 h.

### 4.5. Identification of Mutations in rpoB and rpoC

Resistant colonies were selected on agar with CorA (4 × MIC) employing 19 independent cultures (nine cultures of *S. aureus* HG001 (rifampicin-susceptible), eight cultures of *S. aureus* Mu50 (rifampicin-resistant), and one culture each of *S. aureus* USA300 and *S. aureus* N315 (clinical isolates that are rifampicin-susceptible)). Resistant colonies were picked and purified by streaking on Iso-Sensitest agar (4 × MIC CorA). For the DNA preparation, strains were grown overnight on 4 × MIC CorA-containing agar, and two to three colonies were resuspended in 60 µL of sterile distilled water and heated for 5 min to 100 °C. Cells were removed by centrifugation and the supernatant served as a template for the amplification of *rpoA*, *rpoB*, and *rpoC* (for primers see Table 3). *S. aureus* Mu50 DNA was purified from 1.5 mL of culture using the Molzym Presto-SpinD Bug Kit, including an additional lysis step with 1 mg/mL lysostaphin in RS buffer at 37 °C, 1 h. After agarose gel electrophoresis, bands with PCR products were excised and purified using the Monarch PCR & DNA Cleanup Kit (New England Biolabs, Frankfurt, Germany). Sanger sequencing was performed by Microsynth Seqlab (Göttingen, Germany) using the primers in Table 3. Sequences were evaluated with the Geneious software (Geneious R10 [21]).

## Figures and Tables

**Figure 1 antibiotics-11-00920-f001:**
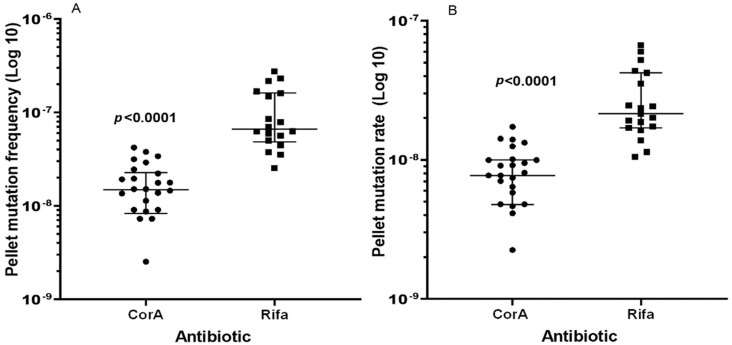
Spontaneous mutations leading to resistance against CorA and Rifa using plated pellets. (**A**) FoM to CorA and Rifa resistance, (**B**) corresponding mutation rates calculated using the Lea–Coulson Method of the Median in FALCOR [14]. Three CorA plates did not show growth and a value of zero cannot be depicted using a logarithmic axis, but these values were included in the calculation of the median values and significance. P-values were computed by the Mann–Whitney-U-Test (GraphPad Prism 9.4 for Windows, GraphPad Prism Software, San Diego, CA, USA). The whiskers indicate 25 and 75 percentiles.

**Figure 2 antibiotics-11-00920-f002:**
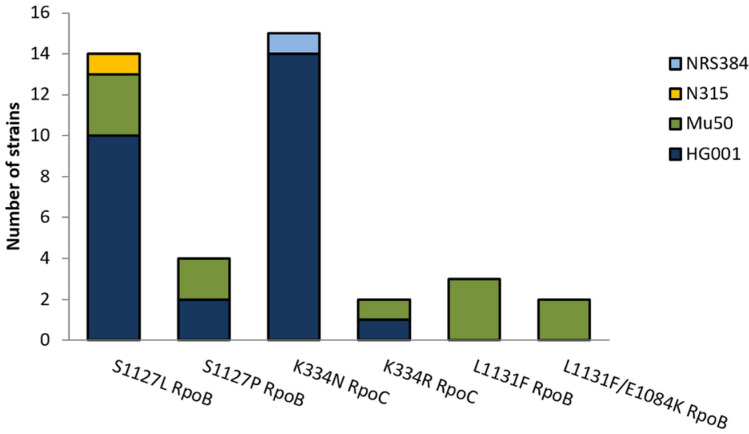
Statistics of amino acid exchanges identified in RpoB and RpoC in 40 CorA-resistant *S. aureus* mutants with different strain and resistance backgrounds (*S. aureus* HG001, N315, Mu50, and NRS384).

**Table 1 antibiotics-11-00920-t001:** CorA MICs for different *S. aureus* strains and their resistant mutants.

Strain	Exchange in Resistant Mutant	CorA MIC
*S. aureus* HG001	Wildtype	0.25 mg/L
*S. aureus* HG001CSz1	K334N RpoC	64 mg/L
*S. aureus* Mu50	Wildtype	0.25 mg/L
*S. aureus* Mu50CSz1	L1131F RpoB	4 mg/L
*S. aureus* Mu50R1.1	S1127P RpoB	32 mg/L
*S. aureus* Mu50R3.1	L1131F/E1084K RpoB	128 mg/L
*S. aureus* Mu50R4.1	S1127L RpoC	64 mg/L
*S. aureus* Mu50R5.1	K334R RpoC	64 mg/L
*S. aureus* N315	Wildtype	0.25 mg/L
*S. aureus* N315CSz1	S1127L RpoB	64 mg/L
*S. aureus* USA300	Wildtype	0.25 mg/L
*S. aureus* USA300CSz1	K334N RpoC	128 mg/L

**Table 2 antibiotics-11-00920-t002:** Frequency of mutations (FoM) leading to resistance and mutation rates of *S. aureus* selected on 4 × MIC CorA or rifampicin using 1 mL or 0.1 mL of culture and two different methods (LC and MSS-MLE) for the calculation of the mutation rates.

	1 mL Culture on CorA Agar (+/− 95% CI ^1^)	1 mL Culture on Rifa Agar (+/− 95% CI)	Ratio Rifa/CorA
Median of FoM	1.48 × 10^−8^ (4.7 × 10^−9^, 5.7 × 10^−9^)	6.67 × 10^−8^ (8.2 × 10^−8^, 1.0 × 10^−8^)	4.5
Mutation rate (LC) ^2^	7.72 × 10^−9^ (2.3 × 10^−9^, 2.9 × 10^−9^)	2.15 × 10^−8^ (1.4 × 10^−8^, 4.1 × 10^−9^)	2.8
Mutation rate (MSS-MLE) ^3^	5.93 × 10^−9^ (2.0 × 10^−9^, 1.79 × 10^−9^)	2.49 × 10^−8^ (7.3 × 10^−9^, 6.6 × 10^−9^)	4.2
	**0.1 mL culture on CorA agar (+/− 95% CI)**	**0.1 mL culture on rifampicin agar (+/− 95% CI)**	**Ratio Rifa/CorA**
Median of FoM	2.27 × 10^−8^ (1.5 × 10^−7^, 2.3 × 10^−7^)	1.27 × 10^−7^ (1.0 × 10^−7^, 4.0 × 10^−8^)	5.6
Mutation rate (LC) ^2^	8.77 × 10^−9^ (5.8 × 10^−9^, 8.8 × 10^−9^)	3.49 × 10^−8^ (2.0 × 10^−8^, 9.0 × 10^−9^)	4.0
Mutation rate (MSS-MLE) ^3^	9.5 × 10^−9^ (3.1 × 10^−9^, 2.8 × 10^−9^)	3.51 × 10^−8^ (8.6 × 10^−9^, 7.8 × 10^−9^)	3.7

^1^ CI: 95% Confidence intervals of the medians (+/−). ^2^ LC: Lea–Coulson Method of the Median. ^3^ MSS-MLE: Ma–Sandri–Sarkar Maximum Likelihood Estimator Method.

**Table 3 antibiotics-11-00920-t003:** Primers used for the PCR amplification and sequencing of RNAP.

Primer	Gene	Sequence	Ref.
rpoAI	*rpoA*	5′-TAACTGCGATCAGAGACGTTACTCC-3′	[17]
rpoAII	*rpoA*	5′-GCTGCATTACGACGAGAAGCTAAAT-3′	[17]
rpoBF	*rpoB*	5′-ATTAGTGTTGCCGTTTTCTTTT-3′	[17]
rpoBR	*rpoB*	5′-AGTATCTTTTTGCCTGTTTTG-3′	[17]
rpoCI	*rpoC*	5′-GACGATGATGTTGTAGAACGCAAAG-3′	[17]
rpoCII	*rpoC*	5′-TGTTGTTTGTTAAAGCGTGCAACTT-3′	[17]
rpoCfor2	*rpoC*	5′-GGACGATTTGCAACAAGTGA-3′	This study
rpoCrev2	*rpoC*	5′-CCAACTGCTTCACCAACTTC-3′	This study
F3	*rpoB*	5′-AGTCTATCACACCTCAACAA-3′	This study
F4	*rpoB*	5′-TAATAGCCGCACCAGAATCA-3′	This study
2955F	*rpoB*	5′-GATCGATATCATGTTAAATCCTCTTGGTGTACCATC-3′	This study
3824R	*rpoB*	5′-CTATTTTCATTAATGGAAATTATTT-ACATCAATCA-3′	This study
955F	*rpoB*	5′-ACTGAAACTGGTGAAATTGTAGTTG-3′	This study
2706R	*rpoB*	5′-GTGAGGTACACGTAATGAAGTATCT-3′	This study
653F	*rpoC*	5′-TTTTGATGTACTTCCAATCATCCCACCAGAAATTC-3′	This study
1552R	*rpoC*	5′-CATATGCTTTTAATACTTCATTTGTATTATTAAAGA-3′	This study

## Data Availability

All data are contained within the article or Appendix A.

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
