# Peer review of "The RNA Polymerase Inhibitor Corallopyronin A Has a Lower Frequency of Resistance Than Rifampicin in Staphylococcus aureus"

_antibiotics, 2022, doi:10.3390/antibiotics11070920_

Round 1
Reviewer 1 Report
The manuscript presented by Balansky et al. describing antimicrobial activity of corallopyronin A against S. aureus strains, represent interesting and useful results. The manuscript is concise, clear and well written. I would like to congratulate the authors, it was a pleasure reading it.
I have just two comments:
Page 6, chapter 4.3 Minimal inhibitory concentration: The authors should state which method was used. I suggest consulting CLSI standards (CLSI-2018-M100-S28, Performance Standards for Antimicrobial Susceptibility Testing).
Page 6, line 227: The authors should clearly specify the concentration of DMSO used as a solvent in the study and include solvent control. Is the percentage of used DMSO appropriate for antimicrobial assays, knowing that the solvent has bactericidal as well as cytotoxic activity for eukaryotic cells?
Author Response
> Thank you for your time and effort spent in reviewing our manuscript and for your constructive comments. All line numbers given below belong to the version with tracked changes.
The manuscript presented by Balansky et al. describing antimicrobial activity of corallopyronin A against S. aureus strains, represent interesting and useful results. The manuscript is concise, clear and well written. I would like to congratulate the authors, it was a pleasure reading it.
I have just two comments:
Page 6, chapter 4.3 Minimal inhibitory concentration: The authors should state which method was used. I suggest consulting CLSI standards (CLSI-2018-M100-S28, Performance Standards for Antimicrobial Susceptibility Testing).
> I have added the EUCAST standard, because we use this method (line 239).
Page 6, line 227: The authors should clearly specify the concentration of DMSO used as a solvent in the study and include solvent control. Is the percentage of used DMSO appropriate for antimicrobial assays, knowing that the solvent has bactericidal as well as cytotoxic activity for eukaryotic cells?
> This was a problem with the resistant mutants, which needed higher CorA concentrations in order to see inhibition. Solvent controls were always included and showed that 25.6 % of DSMO always inhibited bacterial growth. In contrast, 12.8 % of DMSO, which were present in the first well of the MICs of the resistant mutants, were tolerated. This has been added to the manuscript in lines 242-243.
Reviewer 2 Report
The manuscript entitled “The RNA polymerase inhibitor corallopyronin A has a lower frequency of resistance than rifampicin in Staphylococcus aureus” determined the CorA frequency of mutation (FoM) in different S. aureus clinical strains using fluctuation analysis and compared with the results obtained with rifampicin. It contains some very interesting results and is suitable for publication after major revisions.
Lines 84-85: It would be interesting to show the data of this decrease
Lines 90-91: These results are in table 2 and not in table 1
Line 98: Supplementary Appendix Table A1 should be improved (Title, presentation, format, and results)
Line 110-111: Table 1 is confusing and must be reviewed for presentation, layout, and captions. Where is the legend of 1? In the seventh row of the table, the number corresponding to the legend is 3 and not 2
Lines 124-125: Hit the spacing
Lines 125-126: The results of this figure were not explained in the text. Should improve this figure.
Line 227: Write out what DMSO means (dimethyl sulfoxide)
Lines 268-270: Why did the authors employ only one culture of S. aureus USA300 and S. aureus N315?
Line 273: Correct this: “a. dest.”
Line 275: The correct is Table 3, not Table 1
Author Response
> Thank you for your time and effort spent in reviewing our manuscript and for your constructive comments and especially for spotting the incorrect table numbers. All line numbers given below belong to the version with tracked changes.
The manuscript entitled “The RNA polymerase inhibitor corallopyronin A has a lower frequency of resistance than rifampicin in Staphylococcus aureus” determined the CorA frequency of mutation (FoM) in different S. aureus clinical strains using fluctuation analysis and compared with the results obtained with rifampicin. It contains some very interesting results and is suitable for publication after major revisions.
Lines 84-85: It would be interesting to show the data of this decrease.
> The data are shown in Supplemental Fig. A1.
Lines 90-91: These results are in table 2 and not in table 1.
> This mistake has been corrected. The former Table 2 is now Table 1 and has been shifted to the front, since tables should be inserted when they are first mentioned.
Line 98: Supplementary Appendix Table A1 should be improved (Title, presentation, format, and results)
> We have added the title and authors of this paper to the Appendix and a more informative title indicating that this is the raw data set that was used for calculating the values shown in (re-numberes) table 2. Furthermore, the table was separated into two parts, to clearly indicate the method and its format was improved.
Line 110-111: Table 1 is confusing and must be reviewed for presentation, layout, and captions. Where is the legend of 1? In the seventh row of the table, the number corresponding to the legend is 3 and not 2
> The superscripts have been corrected (no 1 has been removed, since it originally referred to a line that had been removed during the modification of the manuscript and the others have been re-numbered). The abbreviation FoM is explained in the header.
Lines 124-125: Hit the spacing
> The spacing between paragraphs has been adjusted to the rest of the text.
Lines 125-126: The results of this figure were not explained in the text. Should improve this figure.
> We have now explained in the text that there were two main mutational hotspots in RpoB and RpoC (lines 146-157), which is demonstrated by the figure. The figure has been improved, including color.
Line 227: Write out what DMSO means (dimethyl sulfoxide)
> done
Lines 268-270: Why did the authors employ only one culture of S. aureus USA300 and S. aureus N315?
> We employed only one culture of these rifampicin susceptible strains, because we had already sequenced 27 CorA resistant colonies from HG001, which also shows susceptibility of rifampicin. We tested more colonies of Mu50 because we were interested in the location of the mutations in a strain with rifampicin resistance. I have indicated this in line 148 ff.
Line 273: Correct this: “a. dest.”
> a. dest. has been replaced with “distilled water”
Line 275: The correct is Table 3, not Table 1
> This has been corrected.
Reviewer 3 Report
The authors have presented their work in an excellent way.
The results are novel and interesting.
The data is presented in an eye catching manner.
The methods are plausible.
The discussion is well written.
Author Response
> Thank you for your time and effort spent in reviewing our manuscript and for your constructive comments.
The authors have presented their work in an excellent way.
The results are novel and interesting.
The data is presented in an eye catching manner.
The methods are plausible.
Reviewer 4 Report
In the current research article entitled " The RNA polymerase inhibitor corallopyronin A has a lower frequency of resistance than rifampicin in Staphylococcus aureus", Bierbaum et al., have investigated and determined the CorA FoM in S. aureus using fluctuation analysis at 4 x minimum inhibitory concentration. They characterized. S. aureus strains HG001, Mu50, N315, and USA300 had an MIC of 0.25 mg/L. Authors have concluded that CorA is a potential antibiotic for treating S. aureus infections. This article addresses a research topic of great interest. However, this reviewer has certain suggestions that would help produce a more comprehensive overview of the topic:
Comments:
1. The English of manuscript can be polished (minor) and there are few typo errors in the manuscript that can be checked.
2. The authors may additionally provide challenges, or prospect of the present study.
3. At least one illustrative figure may be provided as to highlight the summary of this study.
Author Response
> Thank you for your time and effort spent in reviewing our manuscript and for your constructive comments. All line numbers given below belong to the version with tracked changes.
- The English of manuscript can be polished (minor) and there are few typo errors in the manuscript that can be checked.
> The English text has been checked by a native speaker.
- The authors may additionally provide challenges, or prospect of the present study.
> We have added the perspective that CorA will now be tested for its suitably as an antibacterial agent for staphylococcal infections (line 207-209).
- At least one illustrative figure may be provided as to highlight the summary of this study.
> We have provided a graphical abstract, showing the structure of CorA and the amino acids of RpoB and RpoC that interact with CorA and the hotspots and exchanges leading to resistance.